# High-Resolution National-Scale Mapping of Paddy Rice Based on Sentinel-1/2 Data

Chenhao Huang [1,2], Shucheng You [3], Aixia Liu [3], Penghan Li [1,2], Jianhua Zhang [1,2] and Jinsong Deng [1,2,*]

1   College of Environmental and Resource Sciences, Zhejiang University, Hangzhou 310058, China
2   Zhejiang Ecological Civilization Academy, Huzhou 313300, China
3   Land Satellite Remote Sensing Application Center, MNR, Beijing 100048, China
*   Correspondence: jsong_deng@zju.edu.cn

**Abstract:** Rice has always been one of the major food sources for human beings, and the monitoring and planning of cultivation areas to maintain food security and achieve sustainable development is critical for this crop. Traditional manual ground survey methods have been recognized as being laborious, while remote-sensing technology can perform the accurate mapping of paddy rice due to its unique data acquisition capabilities. The recently emerged Google Earth Engine (GEE) cloud-computing platform was found to be capable of storing and computing the resources required for the rapid processing of massive quantities of remote-sensing data, thereby revolutionizing traditional analysis patterns and offering unique advantages for large-scale crop mapping. Since the phenology of paddy rice depends on local climatic conditions, and considering the vast expanse of China with its outstanding geospatial heterogeneity, a zoning strategy was proposed in this study to separate the monsoon climate zone of China into two regions based on the Qinling Mountain–Huaihe River Line (Q-H Line), while discrepant basic data and algorithms have been adopted to separately map mid-season rice nationwide. For the northern regions, optical indices have been calculated based on Sentinel-2 images, growth spectral profiles have been constructed to identify phenological periods, and rice was mapped using One-Class Support Vector Machine (OCSVM); for the southern regions, microwave sequences have been constructed based on Sentinel-1 images, and rice was mapped using Random Forest (RF). By applying this methodological system, mid-season rice at 10 m spatial resolution was mapped on the GEE for the entire Chinese monsoon region in 2021. According to the accuracy evaluation coefficients and publicly released local statistical yearbook data, the relative error of the mapped areas in each province was limited to 10%, and the overall accuracy exceeded 85%. The results could indicate that mid-season rice can be mapped more accurately and efficiently on a China-wide scale with relatively few samples based on the proposed zoning strategy and mapping methods. By adjusting the parameters, the time interval for mapping could also be further extended. The powerful cloud-computing competence of the GEE platform was used to map rice on a large spatial scale, and the results can help governments to ascertain the distribution of mid-season rice across the country in a short-term period, which would be well suited to meeting the increasingly efficient and fine-grained decision-making and management requirements.

**Keywords:** paddy rice mapping; Sentinel-1/2 images; mid-season rice; Google Earth Engine (GEE); national scale; zoning strategy; precision agriculture

## 1. Introduction

As one of the staple foods of human beings, rice accounts for about 12% of the world's arable land and feeds nearly half of the world's population [1]. Thus, the visualization of rice cultivation areas has become a critical task. The traditional mapping of crop cultivation extent requires manual field surveys and step-by-step reporting; these processes are usually time consuming and pose difficulties in obtaining the exact distribution of crops [2,3]. Remote sensing can play a unique role in the monitoring, evaluation, and management

of agricultural production activities in open areas due to its strengths in relation to its expansive coverage, short revisit periods, fast information acquisition, and relatively low cost [4]. In the 21st century, remote-sensing monitoring has become an effective approach for mapping paddy rice due to the combination of high-resolution data and the cross-fertilization of other novel technologies [5].

An in-depth understanding of rice phenology is a non-negligible prerequisite for deciphering the extent of rice cultivation. Although the phenological characteristics exhibited by rice at different growth stages can vary according to geographical environment in which the rice is grown, the genetically determined characteristics of a species cause the growth stages of rice in different locations to follow certain patterns [6–8]. On the other hand, among all food crops, rice is the only one that requires a substantial amount of water at the growth stage, while transplanting is needed [9,10]; therefore, it has become critical to identify rice fields from satellite images enriched with multiple ground objects by examining the dynamic evolution of water–land–vegetation components during an entire growth cycle based on the physiological characteristics of rice during different growth stages.

During the past two decades, remote-sensing-based rice mapping has undergone an evolution from manual interpretation to intelligent monitoring. The earlier visual interpretation methods have been found to be inefficient and costly, while the subsequent machine learning algorithms (e.g., RF and convolutional neural networks (CNNs)) and object-based segmentation algorithms have provided significant improvements [11]. Currently, the recognized features for rice field identification can be classified into two main types: optical vegetation indices and microwave flood signals. Accordingly, two major methodological systems, optics-based mapping and microwave-based mapping, have been developed.

Regarding the optics-based methodology, MODIS, Landsat, and Sentinel-2 data are frequently-used sources [12–17]. To better determine the growth states of rice, researchers usually couple rice phenology with multiple vegetation indices to analyze the profile characteristics of the index time series and establish threshold formulas for pixel-by-pixel rice field extraction [18]. The advantages of this method are its simple principles and ease of use, while its prominent disadvantages are its vulnerability to weather factors (e.g., cloud occlusion) and mixed pixels; therefore, the reliability of the corresponding results depends on the number and quality of images at critical growing stages.

As for the microwave-based methodology, the popular satellite data sources employed are predominantly RadarSat-2, ENVISAT-1, and Sentinel-1 [19–22]. Synthetic aperture radar (SAR) emits microwave signals that can penetrate clouds, theoretically enabling the acquisition of images under various weather conditions. The growth habits of rice dictate that the main grown period is in the rainy season, so SAR data can overcome the disadvantage wherein optical data might be susceptible to cloudy weather [23]. In practice, the time series variation of radar backscatter coefficients during the rice growth phase is generally considered to be the basis for identifying paddy pixels [24,25]. However, insufficient temporal resolution and the speckle noise caused by signal fading have also become essential factors as they limit the application of SAR-based methods [26,27].

Beyond optics-based mapping and microwave-based mapping, the combination of optics and microwaves is a newly developed method that has gradually become a promising class of rice-mapping algorithm in recent years due to technological advances and data expansion. In 2019, Cai et al. mapped rice based on phenology data and time-series Sentinel-2 NDVI and Sentinel-1 SAR images based on an object-based classifier with optimal feature combinations acquired by determining the JBh distance [28]. In the same year, Mansaray et al. explored an approach combining four-source optical satellite (Landsat-8, Sentinel-2A, HJ-1, and Gaofen-1) and microwave satellite (Sentinel-1) images by constructing a spectral index dataset; the approach was used to detect rice fields [29]. In 2020, Fiorillo et al. plotted lowland rice crop areas in the Sédhiou region with an RF algorithm using composite Sentinel-1/2 imagery [30]. The above studies showed that the synergy of optics and microwave images can compensate for the loss of information due to cloudy weather during the critical seasons of paddy rice by incorporating more data and deeper

features, thus achieving a generally better degree of mapping accuracy than that achieved using a single data source. In recent years, with the continuous updating of methodologies, the selection of suitable data combination forms and combination algorithms in order to reduce the impact of cloud pollution has become the key research direction.

Remarkable advances in precision agriculture have continuously driven rice mapping in the direction of adapting to the increasingly efficient decision-making needs of governments [31,32]. The advent of geographic cloud-computing platforms may be capable of providing neoteric solutions for large-scale, accurate, and rapid crop mapping. One of the most iconic methods in this regard is the use of the Google Earth Engine, which was developed by Google in collaboration with Carnegie Mellon University, the National Aeronautics and Space Administration (NASA), and the United States Geological Survey (USGS). To date, it is the world's most advanced cloud-computing platform dedicated to processing remote-sensing imagery and other Earth observation data [33]. With millions of servers worldwide, the GEE is capable of analyzing trillions of images in parallel, leveraging Google's computing infrastructure and open remote-sensing datasets to store more than 40 years of satellite data online, allowing researchers to diagnose near real-time variations in the Earth's surface [34–36]. In contrast to local server operation, the GEE has distinctly improved the computational efficiency of geospatial big data while providing reliable support for processing extensive remote-sensing images. There have been numerous studies conducted regarding crop mapping using the GEE. In 2015, Lemoine et al. proposed the concept of using the GEE to monitor the acreage and growth status of crops globally [37]. In the same year, Lobell et al. selected soybean fields in the Mid-West U.S. as a study area and proposed a scalable crop-mapping method that did not require ground calibration data after fast pre-processing, cloud masking, or calculation of vegetation indices for all Landsat TM/ETM+ data from 2008 to 2013 with the aid of the GEE [38]. In 2019, Liu et al. used Sentinel-2B and Landsat-8 images to extract typical crop phenology and texture features in Tumut Right Banner County, the Inner Mongolia Autonomous Region, based on the GEE platform [39]. All of the above studies highlighted the advantages of GEE mapping quite adequately.

To date, national-scale rice-mapping studies have been conducted in China, Thailand, Malaysia, and other Southeast Asian countries [7,40–42], but their applications have been limited by the coarse spatial resolution (0.5°, 500 m, 250 m, etc.) of the products and the pixel-mixing effect, rendering them incapable of meeting the increasingly fine agricultural management needs in areas with complex topography and smallholder production patterns. Although the GEE may provide a feasible implementation solution for the rapid and accurate identification of rice cultivation ranges at large scales, rice mapping in a huge country like China has still been inadequately studied. Considering the geospatial heterogeneity of rice distribution at the national scale, a zoning strategy for differentiating the monsoon climate zone of China into two regions according to the Q-H Line was proposed. Furthermore, optical data and OCSVM for the northern region, in addition to microwave data and RF for the southern region, with the provincial administrative regions serving as units, have been utilized. As the area of mid-season rice is larger than that of other types of rice, it was selected as the mapping target in this study; thus, a map of the national mid-season rice cultivation extent in 2021, with a high spatial resolution of 10 m, was obtained. The purpose of this study was set to minimize the influence of cloud pollution under the conditions of a shortage of ground-truthing samples while achieving large-scale mapping of rice in the short term and to meet the increasing efficiency and increasingly fine requirements of government departments in administrative management and resource surveys.

## 2. Materials and Methods

### 2.1. Study Area

The People's Republic of China (abbreviated as "China") is located in eastern Asia and on the west coast of the Pacific Ocean (73°33′E–135°05′E, 3°51′N–53°33′N,

Figure 1), possessing a land area of about 9.6 million km$^2$ and a total sea area of about 4.73 million km$^2$ [43]. As the country with the longest history of rice cultivation, China has the world's largest rice acreage and production values and has also been the largest rice consumer [44,45].

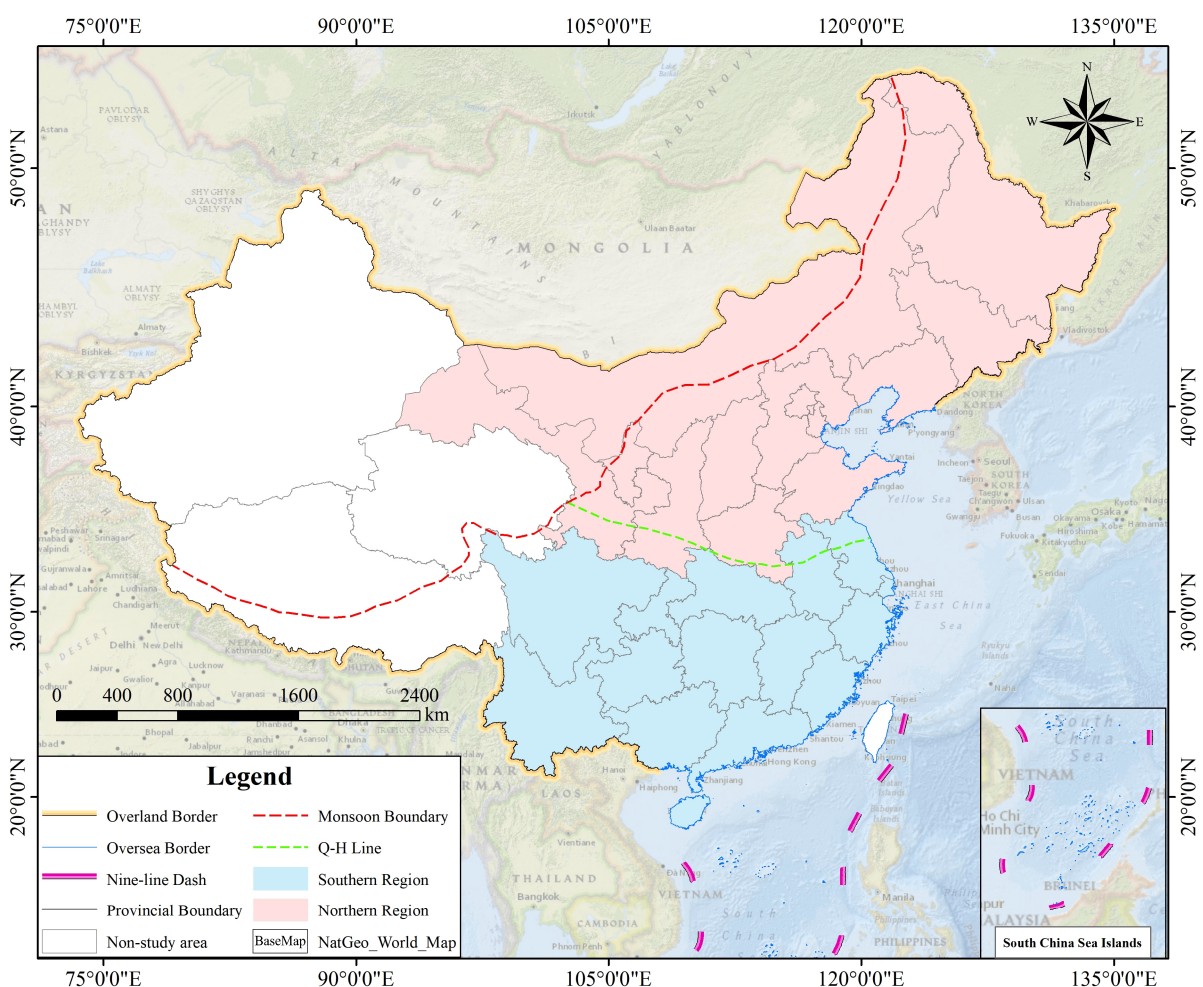

**Figure 1.** Study area (Monsoon region of the People's Republic of China except for Hong Kong, Macao, and Taiwan), geographical boundaries, and zoning status. (Base map layer source: https: //services.arcgisonline.com/arcgis/rest/services, accessed on 4 May 2023.)

Affected by alternating winter and summer winds over a considerable area, China is the country with the most extensive monsoon influence [46]. In geography, the northern boundary of summer wind influence is usually regarded as the dividing line between monsoon and non-monsoon areas. The specific division of monsoon and non-monsoon zones in China is defined by the "Greater Khingan Mountains—Yin Mountains—Helan Mountains—BaYanKaLa Mountains—Kailas Range" line [47]. The areas to the east and south of the line constitute the monsoon zone, and the areas to the west and north of the line constitute the non-monsoon zone. As shown in Table 1, there have been significant differences between the monsoon and non-monsoon climates in China. The high summer temperatures and superior thermal conditions in the monsoon zone of China provide a favorable environment for rice production [48]. Although the southern region (based on the Q-H Line) of China has served as the main area for rice production, it has also been cultivated in many provinces of the northern region. However, there is almost no distribution of rice fields in the non-monsoon region. In order to meet the demand for large-scale rice mapping, the monsoon region of China is addressed in this study.

**Table 1.** Main differences between monsoon and non-monsoon areas in China.

| Area/Characteristic | Monsoon Area | Non-Monsoon Area |
| --- | --- | --- |
| Precipitation | Over 400 mm per year | Below 400 mm per year |
| Relationship between Humidity and Temperature | Synchronous | Out of sync |
| Terrain Type | Plains, basins, and hills | Plateau, mountain, and basin |
| Vegetation Type | Forests and grasslands | Grasslands and deserts |
| River Type | Exoreic rivers mainly recharged by rainwater | Internal streams recharged by melting of snow and ice |
| Agricultural Production Mode | Farming-dominated | Livestock-dominated |

The Qinling Mountain–Huaihe River Line (Q-H Line) is the geographic dividing line between the northern and southern regions of China [49]. To the north and south of this line there are distinct differences in natural conditions (especially precipitation and temperature), geographical landscapes, agricultural practices, and the living habits of people [50]. Accordingly, there have been notable differences in the growth phenology, field size, and distribution density of rice found on both sides of this line. As described in Section 1, based on the performed studies, it was shown that the spectral sequences of rice phenological periods constitute a key feature that distinguishes rice from other crops. In this regard, a zoning strategy dividing China into southern and northern regions based on the Q-H Line (Figure 1) was proposed in this study since the phenological periods of rice are fully dependent on regional climatic conditions and cropping practices.

*2.2. Datasets*

2.2.1. Sentinel-1/2 Data

The optical remote-sensing dataset introduced in this study is the Harmonized Sentinel-2 MSI: Multi-Spectral Instrument, Level-1C; and the microwave SAR dataset is Sentinel-1 SAR GRD: C-band Synthetic Aperture Radar, Ground Range Detected, log scaling. Both datasets were originally provided by the European Space Agency (ESA), are free to access on the GEE platform, and cover the whole year of 2021 and the entire life cycle of all mid-season rice in China. The basic parameters of these two datasets are shown in Table 2.

**Table 2.** Characteristics of the Sentinel series data obtained from the GEE.

| Satellite/Property | Sentinel-1 | Sentinel-2 |
| --- | --- | --- |
| Product Name | Sentinel-1 SAR GRD | Harmonized Sentinel-2 MSI, Level-1C |
| Time Coverage | 1 January–31 December 2021 | 1 January–31 December 2021 |
| Data Unit | dB | Numerical value |
| Spatial Resolution | 10 m | 10–20 m |
| Temporal Resolution | 12 d | 5 d |
| Polarization/Band | VH (IW Mode) | B2 (Blue) B4 (Red) B8 (NIR) B3 (Green) B11, B12 (SWIR) B5-B7, B8A (Red Edge) |
| Pre-processing Steps | Thermal noise removal, Radiometric calibration, Terrain correction | Preliminary cloud removal |

The Sentinel-1 SAR GRD product available on the GEE is a calibrated product that was pre-processed with the Sentinel-1 toolbox (S1TBX), which contains 4 data acquisition modes (SM, IW, EW, and WV) and two polarization bands (VV and VH). The Interferometric Wide (IW) mode and Vertical-transmit/Horizontal-receive (VH) band of this product have been adopted in this study since IW is the main operation mode of this satellite on land that can achieve stable long-term archive retrieval. It was shown that the VH band is more sensitive to rice growth compared to the VV band [23,51,52]. More details about the Sentinel-1 SAR product can be found in the User Guides on the ESA website (https://sentinel.esa.int/web/sentinel/user-guides/sentinel-1-sar/, accessed on 4 May 2023).

Among the publicly released multi-level products of Sentinel-2, the Level-1C product is the Top-of-Atmosphere (TOA) Reflectance product after atmospheric and geometric corrections, and it has eliminated the artifact interference compared to the Level-2A product [53]. In addition, the HARMONIZED dataset from the Sentinel-2 MSI Level-1C product of the GEE employed in this study was de-clouded, and the reflectance band-dependent offset was removed [54]. More details on the Sentinel-2 MSI product can be found in the User Guides on the ESA website (https://sentinel.esa.int/web/sentinel/user-guides/sentinel-2-msi, accessed on 4 May 2023).

### 2.2.2. Sample Data

In this study, a national-scale mapping sample database was established on the Google Earth platform based on visual interpretation methods concerning high-definition satellite images (spatial resolution of 0.54 m), including 7984 rice sample points (Figure 2) and 26,573 non-rice sample points (water bodies, buildings, bare land, forests, grasslands, corn, wheat, cotton, soybean, etc.).

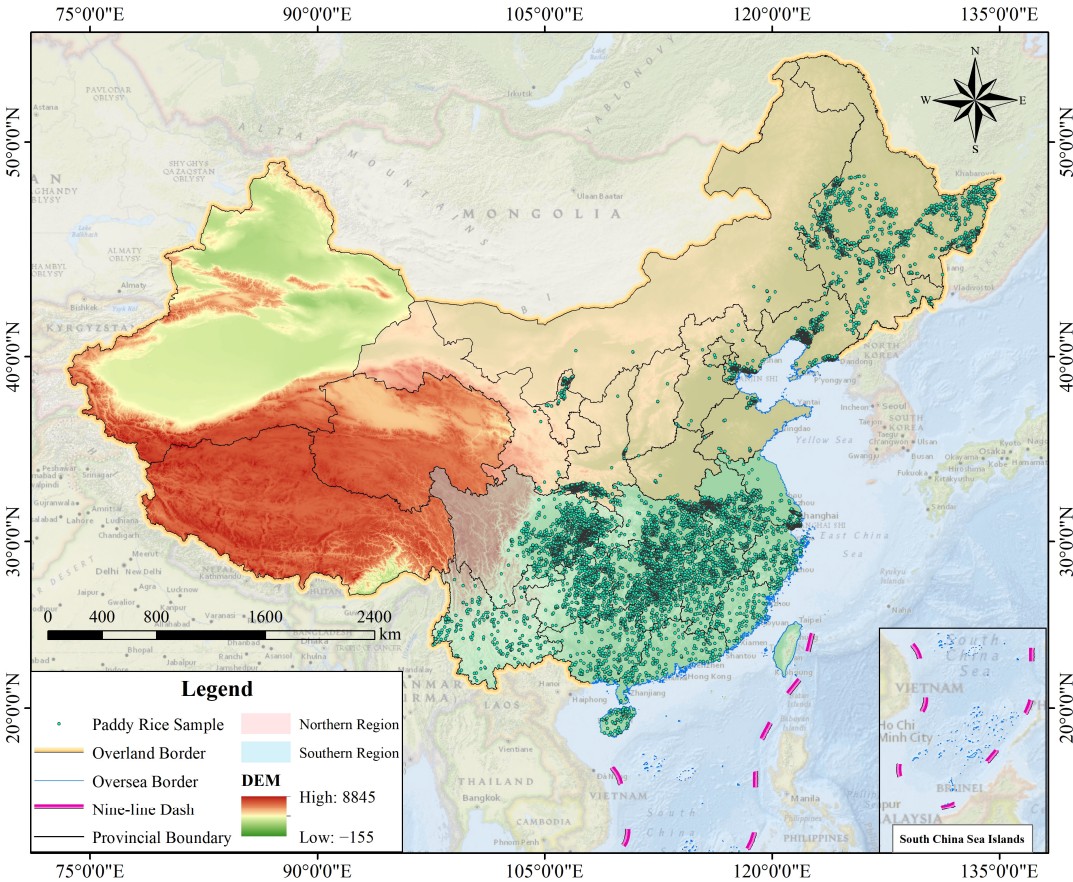

**Figure 2.** Overview of the distribution of mid-season rice samples across China. (Base map layer source: https://services.arcgisonline.com/arcgis/rest/services, accessed on 4 May 2023.)

The steps taken to establish the specific databases used in this study are as follows:

1.  **Topographic masking.** The slope of entire China was calculated using Digital Elevation Model (DEM) data, and areas with slopes $\geq 15°$ were removed via the mask because rice would not be cultivatable within the vast majority of these areas. The DEM data introduced in this study were the NASA SRTM Digital Elevation 30 m data, which can provide worldwide digital elevation values with a spatial resolution of 1 arc-second (~30 m) [55]. The voids in this dataset were filled with the support of other open-source data (ASTER GDEM2, GMTED2010 and NED), which can be downloaded from the GEE platform.

2.  **Reference data overlaying.** Maps of cropping patterns in China during the 2015–2021 period were overlaid on the analyzed area after removing the high-slope mask, and the single-season rice layers were selected and extracted from them. Maps of cropping patterns in China from 2015 to 2021 with a spatial resolution of 500 m have been adopted in this study [56]; they are available on the following open source website (https://figshare.com/articles/dataset/Maps_of_cropping_patterns_in_China_during_2015-2020/14936052, accessed on 4 May 2023).

3.  **Visual interpretation.** The single-season rice layer obtained in step 2 was overlaid onto the Google Earth image layer, and the representative "pure" sample points were selected via manual visual interpretation based on prior knowledge (relating to aspects such as aerial photo comparison and the shape, color, and texture characteristics of the fields) (Figure 3). The correctness of the selected points was examined by dragging the time bar to access the historical images. The total number of sample points in each province and the proportions of their number distributions in each prefecture-level city have been determined according to the proportion of rice planted area in the official statistical yearbooks of each region; i.e., the larger the official statistical rice-sowing area, the more sample points are selected in the provinces and prefecture-level cities. The distribution of the number of rice and non-rice samples according to province can be found in the Supplementary Materials (Table S1).

4.  **Standardized pre-processing.** To standardize the sample database for use in the machine learning model, the sample files (Shapefile format) were imported into the GEE platform and converted into Feature Collection format after point picking conducted on Google Earth. For the rice sample points (i.e., points of interest (POI)), the 'Landcover' attribute was added, and all of the points have been assigned a value of 1. For the non-rice sample points (i.e., non-points of interest (NPOI)), the 'Landcover' attribute has also been added, and all of the points have been assigned a value of 0. Subsequently, each data item in the sample database was buffered in a square area of N m $\times$ N m (the value of N depends on the average size of an independent contiguous paddy field in the provincial administrative region), and the final buffered data (in Feature Collection format) were selected as the sample set required for model training and accuracy verification.

5.  **Random sampling.** The collected rice sample points and non-rice sample points were combined into one dataset and disrupted via random number alignment. A total of 70% of the sample points were randomly selected for model training, while 30% of the sample points were selected for accuracy validation.

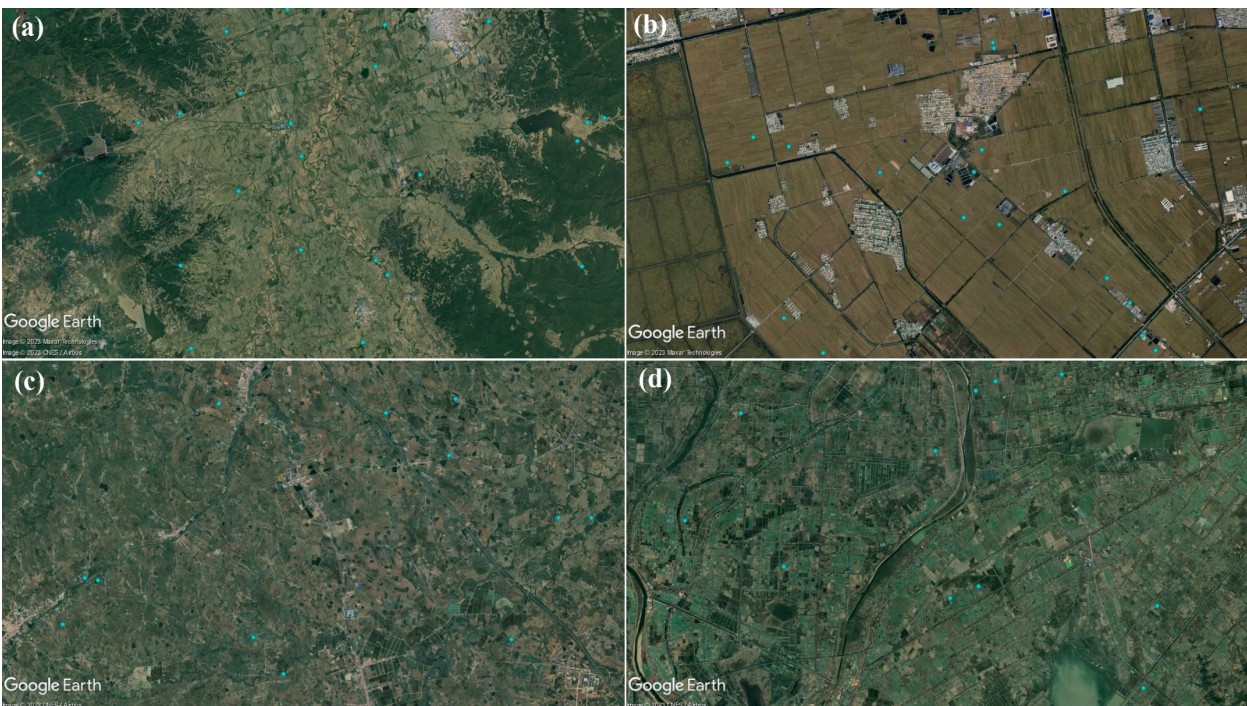

**Figure 3.** Mid-season rice sample collection based on visual interpretation of Google Earth images. (**a**) Partial sample points in Heilongjiang Province (128°24′E, 47°11′N). (**b**) Partial sample points in Liaoning Province (122°12′E, 40°55′N). (**c**) Partial sample points in Anhui Province (116°41′E, 33°20′N). (**d**) Partial sample points in Hunan Province (112°30′E, 28°50′N).

### 2.3. Methodology

The overall technical route of this study (shown in Figure 4) involved 5 steps: zoning strategy development, sample database construction, mid-season rice mapping, post-processing, and accuracy evaluation.

#### 2.3.1. Mapping Methodology for Northern Region

- **Optical feature extraction.**

For rice feature extraction in the northern region, considering that the occurrence of rainy weather in the area to the north of the Q-H Line would be much less than that to the south and that the weather conditions would also be comparatively stable, the information received by the satellite from optical radiation was considered sufficient and capable of being applied to the optical-based mapping method. Aiming to further eliminate the interference of clouds to achieve better mapping accuracy, a de-clouding algorithm and time series median synthesis were implemented in this study. Specifically, first, a de-clouding function was applied to each Sentinel-2 image for the defined phenological period to remove cloudy pixels. Second, considering that clouds move over time, a median synthesis was performed for the images of the specified phenological period (the specific preprocessing code can be found at https://code.earthengine.google.com/c937f24454144ea8efbfc5e9c8 abd396?noload=true, accessed on 2 July 2023). In addition, reliable studies have shown that the vegetation indices calculated based on each waveband of remote-sensing images are essential traits for distinguishing the spectra of target crops from those of other land cover classes [57–60]; therefore, the Bare Soil Index (BSI), Land Surface Water Index (LSWI), Normalized Difference Vegetation Index (NDVI), Plant Senescence Reflectance Index (PSRI), Enhanced Vegetation Index (EVI), and Green Chlorophyll Vegetation Index (GCVI) have been chosen and calculated for all valid pixels within each northern province throughout 2021 based on Sentinel-2 images after de-clouding pre-processing and median synthesis. The formulas for these indices are shown in Table 3 [61].

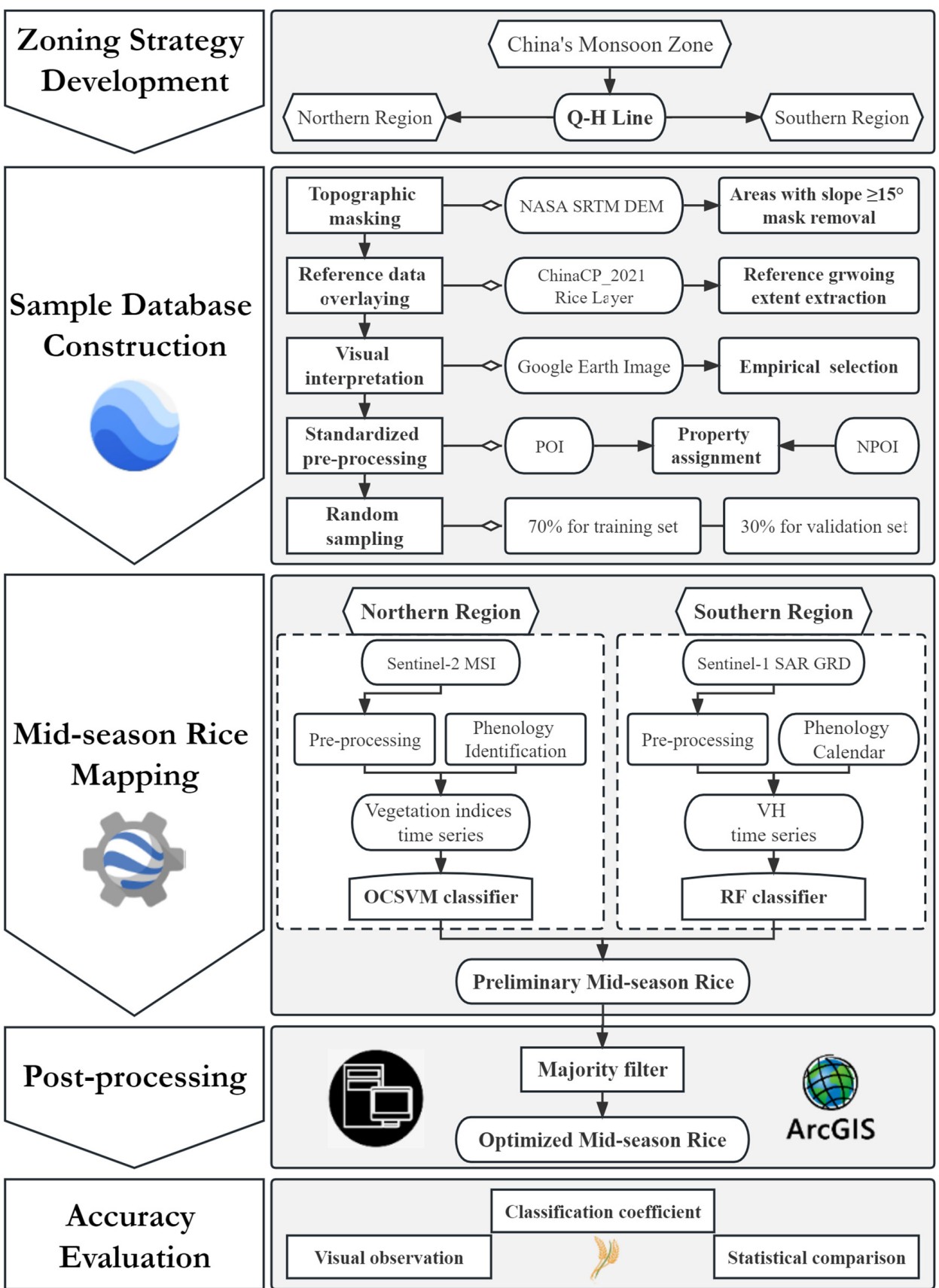

**Figure 4.** Overall technology roadmap.

**Table 3.** Optical indexes based on Sentinel-2 data for rice mapping in the northern region.

| Optical Index | Calculation Formula [1] | Phenological Stages |
|---|---|---|
| Bare Soil Index (BSI) | $BSI = \frac{(SWIR+Red)-(NIR+Blue)}{(SWIR+Red)+(NIR+Blue)}$ | Bare Soil Stage |
| Land Surface Water Index (LSWI) | $LSWI = \frac{NIR-SWIR}{NIR+SWIR}$ | Transplanting Stage |
| Normalized Difference Vegetation Index (NDVI) | $NDVI = \frac{NIR-Red}{NIR+Red}$ | Growing Stage |
| Plant Senescence Reflectance Index (PSRI) | $PSRI = \frac{Red-Blue}{RedEdge}$ | Mature Stage |
| Enhanced Vegetation Index (EVI) | $EVI = \frac{2.5\times(NIR-Red)}{(NIR+6\times Red-7.5\times Blue+1)}$ | Growing Stage |
| Green Chlorophyll Vegetation Index (GCVI) | $GCVI = \frac{NIR}{Green} - 1$ | Transplanting Stage |

[1] Red, Blue, Green, NIR, SWIR, and Red Edge indicate the reflectance of red band, blue band, green band, near-infrared band, short-wavelength infrared band, and red edge band, respectively.

- **Phenological stage division.**

For the phenological staging in the northern region, a pixel-based phenological feature composite method (Eppf-CM) proposed by Ni et al. was referenced to determine four major phenological stages of rice growth (Bare Soil Stage, Transplanting Stage, Growing Stage, and Mature Stage) [61]. In this study, it is shown that this method can enhance the spectral separability of paddy rice from other crops and make efficient use of information from multiple representative phenological periods of paddy rice to characterize the spectral profiles during the entire life cycle. By analyzing the time series profiles of the mean values of BSI, LSWI, NDVI, and PSRI for mid-season rice sample sites in the northern provinces, four different critical phenological periods were delineated. Specifically, for the bare soil stage, BSI detection was employed since it can effectively characterize the mineral content of soil, with a persistently high BSI and relatively low NDVI and LSWI. For the transplanting stage, LSWI was chosen since numerous studies have shown its excellent performance with respect to identifying flood signals with an LSWI rising rapidly and a BSI falling. For the growing stage, NDVI was chosen as it is a well-recognized vegetation sensitivity index, whose value increased and peaked during this period, with a decreasing BSI. For the mature stage, PSRI was adopted because of its high responsiveness to changes in carotenoid and chlorophyll content, with PSRI rising significantly and NDVI decreasing.

To efficiently evaluate the dynamics of the four optical indices at a given sample point to help better delineate the rice phenological periods, a script for viewing the time series profiles of BSI, LSWI, NDVI, and PSRI at the sample location for 2021 based on the Inspector function of the GEE was formulated. Users can simply run the script to load the images and select the corresponding rice sample points with Inspector to view the index profile from the window on the right side of the interface. The four optical index profiles for a rice sample point in Liaoning Province are shown in Figure 5. The code of this GEE script can be accessed via the following link (https://code.earthengine.google.com/c816a64f1307576 bbf6de2b9b217f217, accessed on 4 May 2023).

- **Optical index sequence construction.**

Previous studies have shown that EVI is insensitive to soil background and aerosol responses, which are less likely to be saturated in areas with high vegetation cover, and performs better than NDVI in environments with higher humidity [62,63]. Moreover, GCVI can reflect chlorophyll content in leaves through the reflection ratio of near-infrared and green light bands, thus allowing for the accurate determination of the growth dynamics of leaves during transplanting stages [64,65]. Therefore, in order to enhance the identification of the two time periods with prominent growth characteristics, the EVI and GCVI have been calculated separately according to the equation given in Table 3 for each image acquired based on the results of the weather period classification scheme mentioned above. The corresponding time series have also been obtained. Finally, a synthesis operation was performed for the respective phenological periods along with the four optical index

time series already calculated in the above step. The six index sequences were then superimposed via band fusion to produce a new feature image for subsequent machine learning model training.

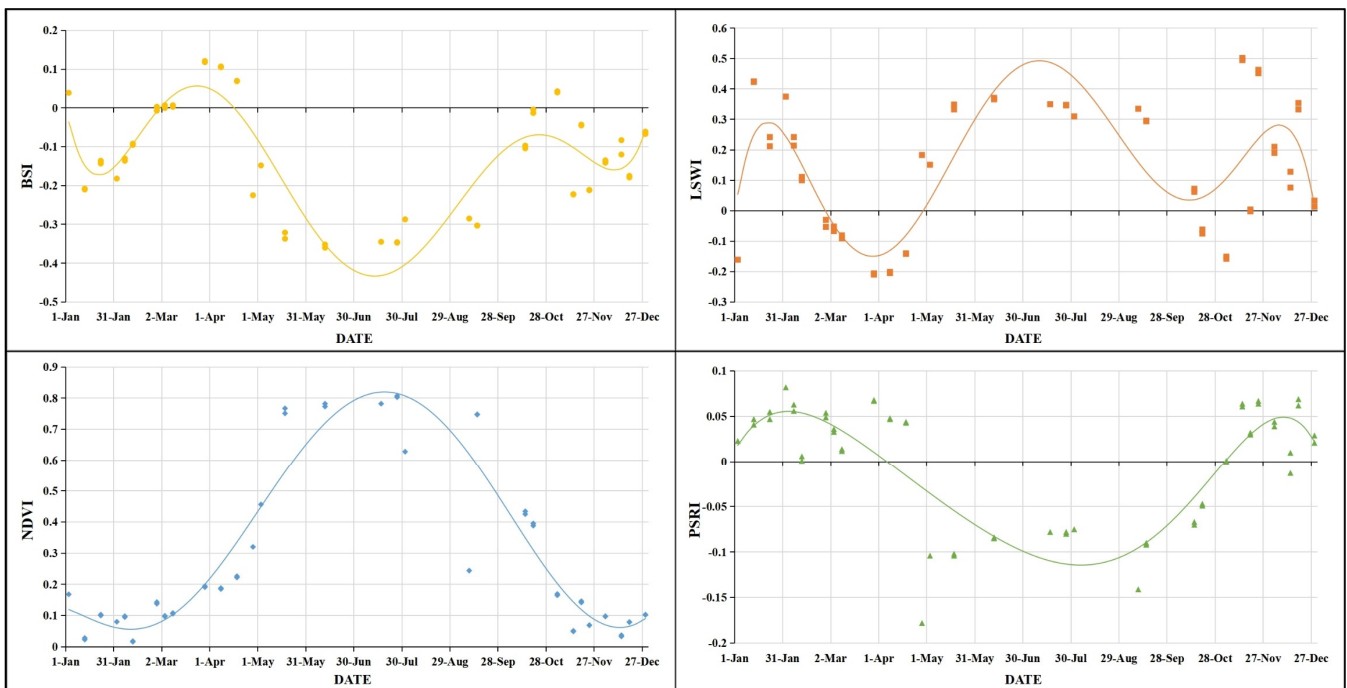

**Figure 5.** Four optical index profiles of a mid-season rice sample point (122°12′E, 40°55′N) in Liaoning Province in 2021. (The four curves in the figure are sixth-order polynomial fitted trend lines. Missing values mean that there is no valid pixel and the vegetation index is not available for calculation at that time.)

- **Machine learning algorithm—OCSVM.**

The machine learning algorithm utilized for rice mapping in the northern region is One-Class Support Vector Machine (OCSVM), which is one of the built-in classifiers of the GEE [66]. This algorithm is a classical anomaly detection algorithm, and its principle is similar to that of a SVM: it assumes that there are two classes of points in the sample space, and it was designed to find a division hyperplane to separate these two classes of samples; the division hyperplane should choose the one with the best generalization ability [67]. Unlike SVM, which is concerned with the binary classification problem, OCSVM has only one classification of interest, which exactly meets the need for anomaly detection, and usually does not focus on anomalous data [68]. Compared with traditional anomaly detection methods, OCSVM can more adequately handle problems containing high-dimensional data, complex datasets, noisy data, and missing data [69].

In addition, it can be empirically gleaned that the rice fields in the northern region are highly contiguous, while the individual field sizes are larger than in the southern regions. Therefore, the sample buffer size (N value) was uniformly set to 10 m for each northern province. Furthermore, 10 m × 10 m buffers were generated for each sample/non-sample point, and the feature images output by the Eppf-CM were then randomly sampled in these buffers and inputted to the OCSVM model. Since these points have been randomly distributed over the entire region of the target ROI, it can be surmised that they adequately represent the variation in spectral information. All the model-training, pixel-based classification, and accuracy validation procedures were performed using the GEE.

### 2.3.2. Mapping Methodology for Southern Region

- **Microwave feature extraction.**

Experiential knowledge has suggested that the range and trend of VH backscatter are promising features with which to distinguish rice from other types of land cover [70,71]. This was demonstrated by the low value of VH in rice pixels during the transplanting stage due to the inundation of water (Figure 6b). Subsequently, due to the growth of rice, the volume scattering of the plants gradually improved, and the backscattering increased significantly during the growing stage. Finally, the backscattering decreased due to the surface scattering from the bare land that remained after the rice had matured and been harvested [72]. Therefore, this "V"-shaped falling–rising feature of VH is the key factor for rice field identification in the cloudy southern region. To visually represent this feature, a script to view the VH backscattering time series profile at the target site over 2021 was developed on the GEE. Figure 6 shows the VH backscatter time series profiles for various land cover classes including mid-season rice in a region of Anhui Province, where the typical "V" shaped fluctuations can be seen in the red circle in Figure 6a; the figure also shows that the time coincided with the local rice-transplanting stage. The code for this GEE script can be accessed via the following link (https://code.earthengine.google.com/6f95a8 a252182c4d736bcaae47bf2024, accessed on 4 May 2023).

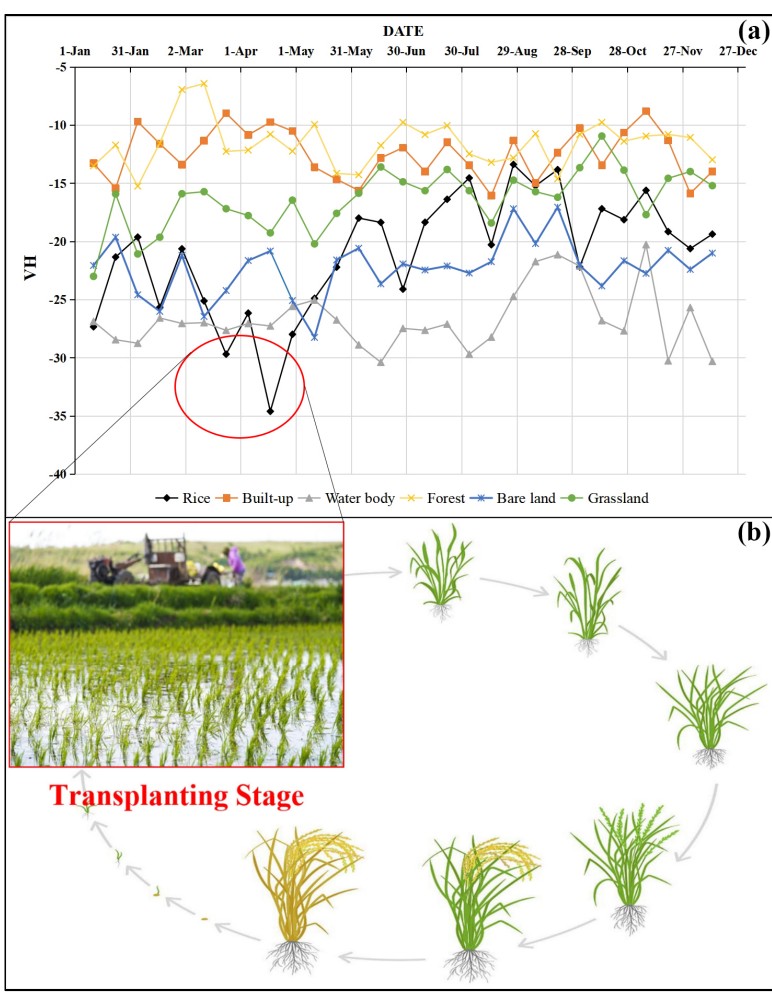

**Figure 6.** Extraction of phenological characteristics of mid-season rice in southern provinces. (**a**) Time-series curves of VH polarization backscatter coefficients of various land classes of Sentinel I in a region of Anhui Province (116°55′E, 33°25′N) in 2021. (**b**) Growth patterns of rice at various phenological stages.

- **Phenological stage division.**

The complex topography, variable climate, and geographically distinctive water and heat conditions in the southern region resulted in the prominent spatial heterogeneity of rice phenology, and an analysis conducted based on characteristic time series profiles would no longer be effective. And there has no study been found which could identify rice phenology accurately and precisely for the entire southern region of China. Therefore, this study referred to the publicly available calendar of rice phenology information on the official website of the Ministry of Agriculture and Rural Affairs of the People's Republic of China (Table 4). This calendar provides multi-stage phenological information on the growth of the main crops throughout China in a geographically zoned manner with a decadal (near 10 d) time scale.

**Table 4.** Rice phenology in southern region of China in 2021 [1].

| Region/Stage | Jianghuai [2] | Jiangnan [3] | Huanan [4] | Xinan [5] |
|---|---|---|---|---|
| Sowing Stage | - [6] | 4.1–4.30 | 1.21–2.20 | 3.11–3.20 |
| Seedling Stage | 4.11–5.31 | 5.1–5.31 | 2.21–3.20 | 3.21–3.31 |
| Transplanting Stage | 6.1–6.10 | 6.1–6.10 | 4.1–4.20 | 4.1–5.20 |
| Greening Stage | 6.11–6.20 | 6.11–6.20 | - | - |
| Tillering Stage | 6.21–7.31 | 6.21–7.31 | 4.21–5.31 | 5.21–7.10 |
| Booting Stage | 8.1–8.10 | - | 6.1–6.10 | 7.11–7.31 |
| Heading Stage | 8.11–8.20 | 8.1–8.10 | 6.11–6.20 | 8.1–8.10 |
| Grouting Stage | 8.21–8.31 | 8.11–8.20 | - | 8.11–8.20 |
| Milk-ripe Stage | 9.1–9.10 | 8.21–10.31 | 6.21–6.30 | 8.21–9.20 |
| Mature Stage | 9.11–9.20 | - | 7.11–7.31 | 9.21–10.20 |

[1] From the Crop Information Calendar on the website of the Ministry of Agriculture and Rural Affairs of the People's Republic of China (http://zdscxx.moa.gov.cn:8080/nyb/pc/calendar.jsp, accessed on 4 May 2023). [2] Including Anhui, Hubei, and Jiangsu. [3] Including Zhejiang, Shanghai, Jiangxi, Hunan, Fujian, and Taiwan. [4] Including Guangdong, Guangxi, Hainan, Hong Kong, and Macau. [5] Including Chongqing, Sichuan, Guizhou, and Yunnan. [6] "-" indicates that this phenological period does not exist for rice in the region or is too short to be measured.

- **VH sequence construction.**

The prerequisite step for conducting rice extraction in the southern region is to construct SAR image sequences based on microwave images and phenological periods for pixel-by-pixel classification. The specific process of constructing SAR image sequences is as follows: Firstly, the Sentinel-1 images should be screened on the GEE according to the phenological calendar. Secondly, the median values of the images in each corresponding phenological period should be synthesized individually, while the mosaic operation should be applied to multiple images at the same location in the specified area. Finally, the synthesized images should be the SAR image units per phenological period, which are then used for the subsequent training of the machine learning model.

- **Machine learning algorithm—RF.**

The machine learning algorithm that was utilized for rice mapping in the southern region is Random Forest (RF). RF is a classifier that contains multiple decision trees [73]. It can generate a new set of training samples by repeatedly randomly selecting n samples from the original training sample set through a bootstrap resampling technique; then, it generates m decision trees to form a random forest, and the classification result of the new data is determined by the number of votes formed by the classification trees [74,75]. In essence, RF is a modification of the decision tree algorithm wherein multiple decision trees are combined, and the establishment of each tree depends on independently drawn samples. Many relevant studies have shown that the RF algorithm presents high classification accuracy

and can successfully handle high data dimensions, thus indicating its high computational speed and robustness toward overfitting [28,76,77].

In addition, our preliminary observations of the images have revealed that the distribution pattern of the rice fields in the southern region varies greatly, presenting both large fields with high contiguity and small, fragmented, and scattered fields; therefore, the sample buffer size (N value) for each southern province was optimized according to the actual situation of different places after several experiments. After N m × N m buffers were generated for each sample/non-sample point, the constructed SAR sequence images were randomly sampled in these buffers and inputted to the RF model (the number of trees in the RF model was set to 500, and the other parameters were set to the default values).

### 2.3.3. Post-Processing Method

Once the preliminary results (in GeoTIFF format) were obtained, two important challenges arose at the same time, which could not be ignored: the impulse noise inherent in the Sentinel series data and the hole phenomenon caused by the lack of consideration of spatial correlation in pixel-by-pixel classification methods [18,61]. Thus, the extracted maps have been imported to local PCs, and ArcGIS software was introduced to perform spatial filtering operations in order to produce more reliable classification results. These operations were incorporated to alleviate noise, repair holes, and minimize misclassification errors at neighboring edges of different features. Specifically, the extraction results have been optimized using the Majority Filter (MF) in the ArcMap10.7 toolbox.

The basic principle of MF is to replace the pixels in the raster based on the mode of neighboring values (Figure 7) [78]. The optional parameters of this tool include "number_neighbors" (which determines the number of neighboring pixels in the filter kernel, whose default value is "FOUR", which means that the kernel of the filter will be the four direct (orthogonal) neighbors to the present cell) and "majority_definition" (which specifies the number of neighboring pixels that must have the same value, the default value of which is "MAJORITY", which means that a majority of cells must have the same value and be contiguous). In this study, the two parameters have been set as default values.

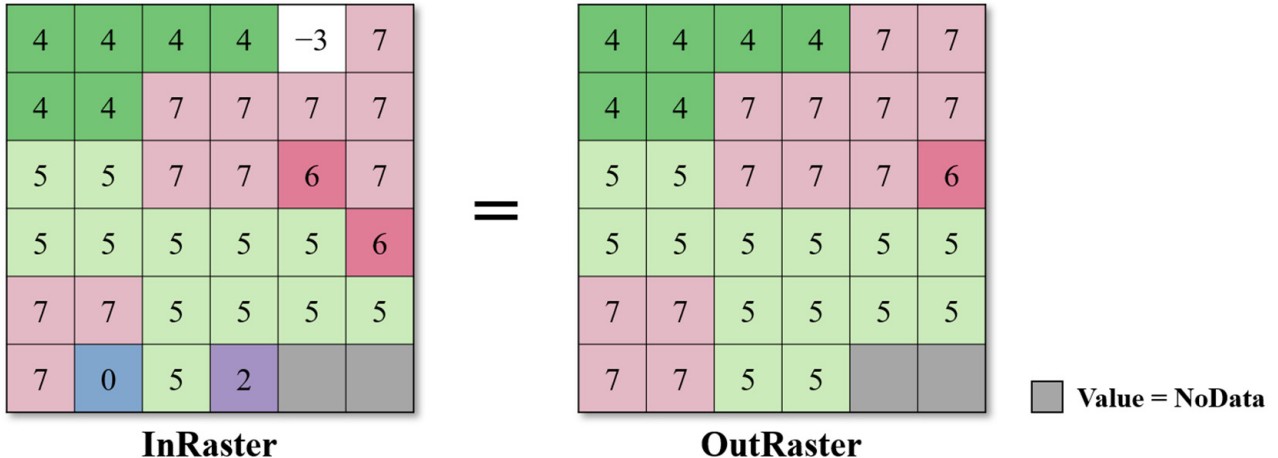

**Figure 7.** Schematic diagram of the process of majority filtering.

As shown in Figure 8, the output raster was stabilized, some of the scattered misclassified rice field pixels were removed, and fine voids within some of the contiguous rice field patches were filled after running the plurality filter several times, while the result would then be selected as the final outcome of rice mapping.

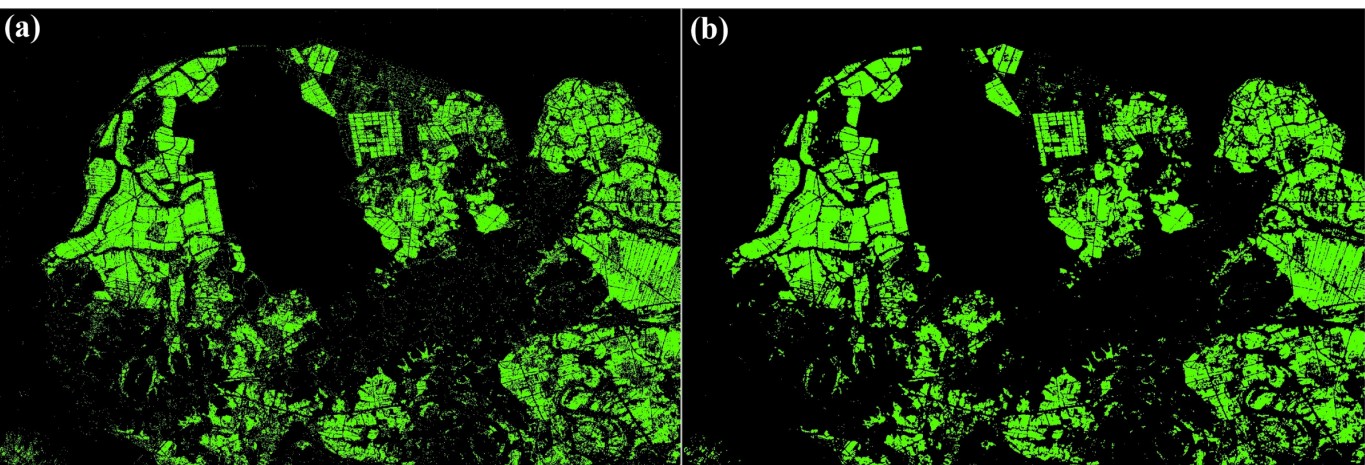

**Figure 8.** Comparison of mid-season rice-mapping results in a certain district of Jiangxi (116°31′E, 29°8′N) before and after the process of majority filtering. (**a**) Before filtering. (**b**) After filtering.

### 2.3.4. Mapping Accuracy Validation

Due to the lack of publicly available China-wide high-resolution (≤10 m) mid-season rice-tagging data, it was not feasible to reliably evaluate the accuracy of the mapping results at the individual pixel scale. Therefore, the visual observation method, the confusion matrix-based classification coefficient method, and the statistical yearbook area comparison method have been carried out for multi-perspective accuracy verification of the results in this study.

- **Visual observation method.** Visual observation is a direct method for evaluating mapping performance. It involves overlaying classification results onto high-definition remote-sensing reference images from Google Earth and matching them.
- **Classification coefficient method.** By invoking the built-in confusion matrix calculation function of the GEE, various accuracy evaluation coefficients can be quantified, such as Producer's Accuracy (PA), User 's Accuracy (CA), Overall Accuracy (OA), Kappa Coefficient (KC), etc., and the results can be quantitatively tested. The specific formulas for the metrics mentioned above can be found in the Supplementary Materials (Table S2).
- **Statistical yearbook comparison method.** The statistical area comparison method is designed to evaluate the classification results from the perspective of quantitative statistics by counting the number of rice pixels within the mapping results of each province before converting them into rice field areas according to the raster size and then comparing them with the rice-sowing areas noted in the statistical yearbooks of each province to acquire the relative errors. The statistical yearbooks of each province can be found on the official websites of the statistical offices of the local governments.

## 3. Results

### *3.1. Mapping Results for Northern Region*

For the northern region, three northeastern provinces (Heilongjiang, Jilin, and Liaoning), which have been highly representative and have the highest statistical area of rice cultivation among the northern provinces, have been selected for the presentation of the mid-season rice-mapping results and accuracy evaluation due to the space limitation of the paper.

### 3.1.1. Overall Distribution

The overall distribution of mid-season rice in the three northeastern provinces of China in 2021 is illustrated in Figure 9. Among these provinces, mid-season rice in Heilongjiang Province (Figure 9a) presented the widest distribution and the highest planting density and

contiguity, which was spatially concentrated in the central and eastern flat terrain areas and broadly distributed in the west. The distribution of rice in the south turned out to be relatively small, and there are almost no large rice areas in the northern mountainous regions. In addition, most of the mid-season rice in Heilongjiang was observed to be distributed along rivers, which could be related to the water-loving characteristic of rice. Most of the mid-season rice in Jilin Province (Figure 9b) is distributed in the northwestern and central parts, and the field contiguity was determined to be the lowest among the three provinces, which is consistent with the topographic characteristics of Jilin Province (higher in the southeast and lower in the northwest). In Liaoning Province (Figure 9c), the topography was observed to be higher in the east and west and lower in the middle; therefore, most of the mid-season rice distribution is concentrated in the middle, which is the most morphologically concentrated but smallest area among the three provinces.

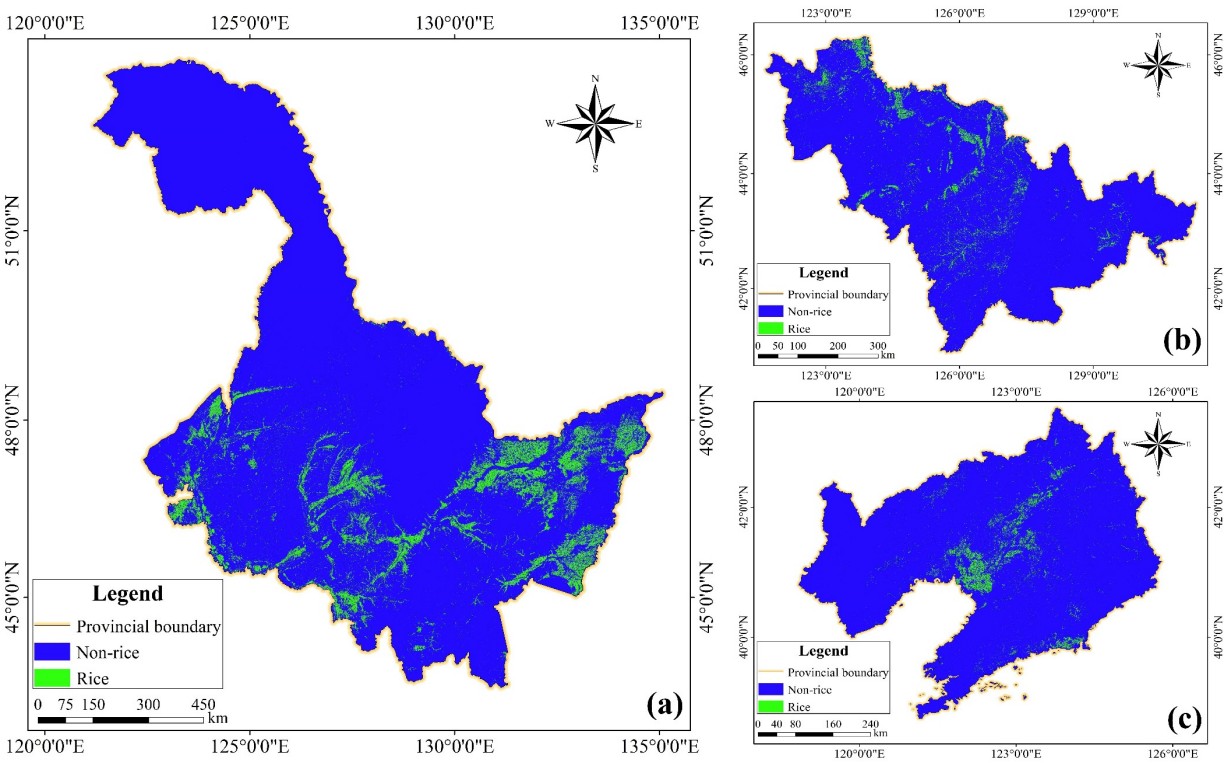

**Figure 9.** Overall distribution of mid-season rice cultivation areas of three typical provinces in northern China in 2021. (**a**) Heilongjiang Province. (**b**) Liaoning Province. (**c**) Jilin Province.

### 3.1.2. Local Visual Comparison

The partial results of rice mapping in the three northeastern provinces and their comparison with real high-definition images are shown in Figure 10. The visual matching of the three provinces led to a largely consistent discovery: the mid-season rice fields in the northeast have been found to be more regular in shape, larger in size, and higher in contiguity. The optical index mapping method based on Sentinel-2 data and the OCSVM algorithm showed excellent visual performance and could clearly distinguish mid-season rice from other land categories (buildings, water bodies, bare land, and other crops). The model extraction resulted in clear edge segmentation and the clear identification of narrow roads and bridges crisscrossing the field. However, many tiny holes were detected inside some of the individual contiguous paddy patches when zooming in for closer observation. These holes somehow matched the distribution pattern of noise in the Sentinel-2 data.

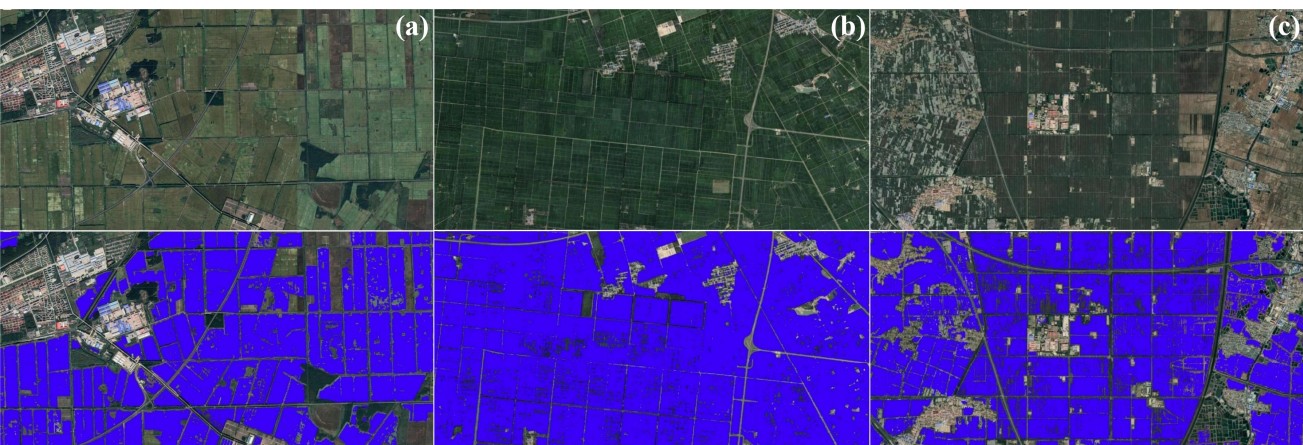

**Figure 10.** Partial results of mapping the cultivation extent of mid-season rice in 2021 in three typical provinces in northern China against high-definition Google Earth satellite imagery. The blue blocks represent the extracted mid-season rice fields. (**a**) Heilongjiang Province. (**b**) Jilin Province. (**c**) Liaoning Province.

### 3.1.3. Accuracy Evaluation

The accuracy of the results of mid-season rice mapping in the three northeastern provinces based on classification coefficients and statistical yearbook areas is reported in Table 5. With a uniform buffer size of 10 m, the extraction results of the three northeastern provinces showed satisfactory classification accuracy. The OA in all three provinces exceeded 90%, while the KC values exceeded 85%. On the other hand, the relative errors with the rice sown area data in the official statistical yearbook were limited to 4%.

**Table 5.** Accuracy evaluation of rice-mapping results in three typical northern provinces.

| Province/ Index | Heilongjiang | Jilin | Liaoning |
|---|---|---|---|
| Buffer Size N (m) | 10 | 10 | 10 |
| Confusion Matrix | [62, 2], [0, 43] | [55, 7], [0, 38] | [43, 5], [0, 29] |
| User's Accuracy (UA) | 0.9556 | 0.8444 | 0.8529 |
| Producer's Accuracy (PA) | 0.9688 | 0.8871 | 0.8958 |
| Overall Accuracy (OA) | 0.98 | 0.93 | 0.94 |
| Kappa Coefficient (KC) | 0.96 | 0.87 | 0.87 |
| Mapping Result Area (km$^2$) | 37,345.73 | 8139.26 | 5009.72 |
| Reference Area (km$^2$) | 38,670 | 8373 | 5206 |
| Relative Error (%) | −3.42 | −2.79 | −3.77 |

### 3.2. Mapping Results for Southern Region

For the southern region, seven provinces (Hunan, Hubei, Jiangxi, Anhui, Jiangsu, Zhejiang, and Guangxi) with highly representative and statistically large areas of rice cultivation have been selected for the presentation of mid-season rice-mapping results and accuracy evaluation due to the space limitation of the paper.

### 3.2.1. Overall Distribution

The general distribution of mid-season rice in seven provinces in southern China in 2021 is shown in Figure 11. The pattern of divergence in rice distribution with variations in topography, temperature, and hydrology is clearly reflected. It is noticeable that the distribution of mid-season rice in Hunan Province (Figure 11a) is widespread and dispersed

with cultivation in all corners of the province, and the relatively most-concentrated area is located in and around the Dongting Lake basin. In Anhui Province (Figure 11b), mid-season rice varied greatly from north to south, with high planting density in the Huaihe Plain in the north where continuous rice fields have surrounded almost all cities and rivers, and scattered rice fields were found in the Jianghuai Tableland hills in the south. A relatively vast area of mid-season rice was observed in Hubei Province (Figure 11c), which has mostly clustered in the central lowland area surrounded by mountains. Jiangsu Province (Figure 11d) was identified as the most ideal area for the growth of rice as it is the province with the highest proportion of plain terrain (86.89%) nationwide and it has a sub-tropical monsoon climate; therefore, mid-season rice is widely distributed. Most of the mid-season rice in Jiangxi Province (Figure 11e) is concentrated in and around the Poyang Lake basin in the north, with a scattered presence in the mountainous hilly areas in the south. In Zhejiang Province (Figure 11f), the geomorphological characteristics can be summarized as "seven mountains, one water and two fields"; thus, the rice cultivation area was found to be relatively small, and most of it is clustered in the Hangjiahu Plain in the northeast. In the Guangxi Zhuang Autonomous Region (Figure 11g), the vast low mountains and hills precipitated a complex distribution of mid-season rice, which was mostly found in river valleys.

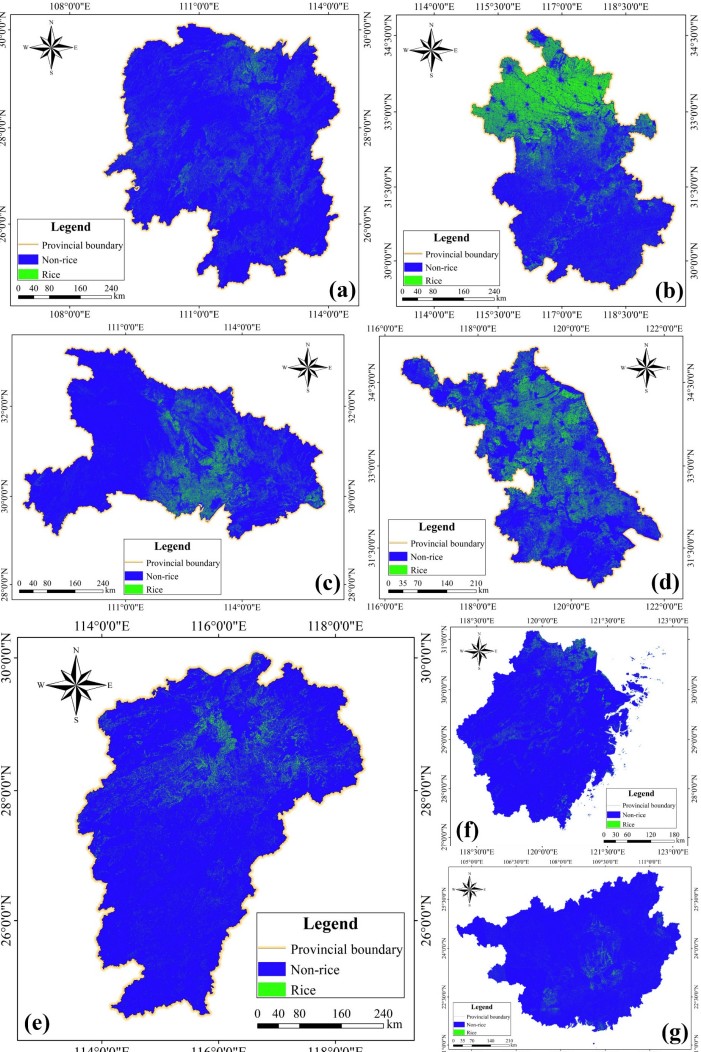

**Figure 11.** Overall distribution of mid-season rice cultivation areas of seven typical provinces in southern China in 2021. (**a**) Hunan Province. (**b**) Anhui Province. (**c**) Hubei Province. (**d**) Jiangsu Province. (**e**) Jiangxi Province. (**f**) Zhejiang Province. (**g**) Guangxi Zhuang Autonomous Region.

### 3.2.2. Local Visual Comparison

The partial results of rice mapping in the seven southern provinces and their comparison with the real high-definition images are shown in Figure 12. It can be found that compared with the northern region, the mid-season rice fields in the south have fragmented shapes and strong spatial heterogeneity of field sizes. The SAR sequence-mapping method based on Sentinel-1 data and the RF algorithm provided relatively effective differentiation of land classes, especially the rejection of water bodies. It is significant that the mapping procedure also showed many misclassifications, i.e., buildings, water bodies, trees, and other crops were misclassified as rice, or rice was identified as other land classes. Some irregularly randomly distributed multi-scale patches were also detected in the results. The uncertainties in classification caused by these phenomena were most evident in provinces with a complex topography and a small statistical area of rice, such as Zhejiang (Figure 12f) and Guangxi (Figure 12g).

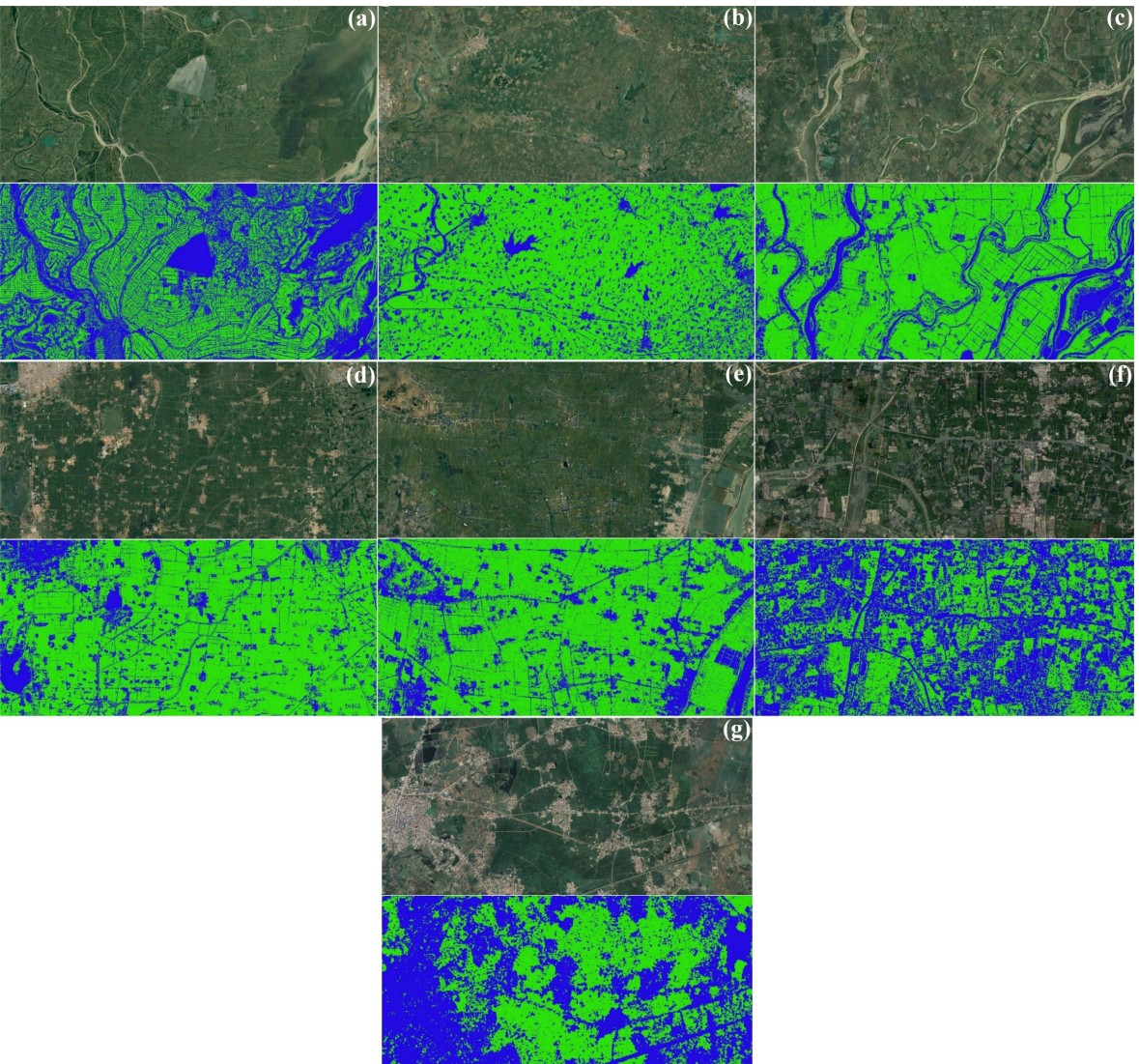

**Figure 12.** Partial results of mapping the cultivation extent of mid-season rice in 2021 in seven typical provinces in southern China against high-definition Google Earth satellite imagery. The green blocks represent the extracted mid-season rice fields, and the blue blocks represent non-rice fields. (**a**) Hunan Province. (**b**) Hubei Province. (**c**) Jiangxi Province. (**d**) Jiangsu Province. (**e**) Anhui Province. (**f**) Zhejiang Province. (**g**) Guangxi Zhuang Autonomous Region.

### 3.2.3. Accuracy Evaluation

The accuracy of the results of mid-season rice mapping based on classification coefficients and statistical yearbook areas in the seven southern provinces are reported in Table 6. As mentioned above, the average size of rice fields varied substantially from province to province in the southern region; therefore, the relatively optimal buffer size (N value) was determined after multiple trials in each province in this study. The OA of all seven southern provinces exceeded 85%, while the KC values exceeded 70% and the F-scores were ≥ 75%; in the meantime, the relative errors in the rice sowing area data in the official statistical yearbook were limited to 10% and lower after majority filtering. With regard to this bias, some studies have found that a large number of rice fields might be abandoned while the paddy fields are converted to dry land (i.e., planted with other crops or vegetables) in areas with complex topography after field investigations due to the difficulty of implementing mechanized rice cultivation [11]. Regarding the official village-based statistics on rice cultivation area, many farmers might not have reported the actual rice cultivation area of their land, which could be the main factor leading to the deviation between the estimation of the statistical and remote-sensing data [79,80].

**Table 6.** Accuracy evaluation of rice-mapping results in seven typical southern provinces.

| Province | Hunan | Hubei | Jiangxi | Jiangsu | Anhui | Zhejiang | Guangxi |
|---|---|---|---|---|---|---|---|
| **Buffer Size N (m)** | 3 | 5 | 4 | 5 | 5 | 3 | 2 |
| **Confusion Matrix** | [68, 22], [4, 67] | [94, 12], [12, 72] | [62, 16], [9, 51] | [93, 7], [5, 77] | [100, 6], [5, 38] | [80, 7], [10, 44] | [104, 3], [6, 22] |
| **User's Accuracy (UA)** | [0.94, 0.75] | [0.89, 0.86] | [0.87, 0.76] | [0.95, 0.92] | [0.95, 0.86] | [0.89, 0.86] | [0.95, 0.88] |
| **Producer's Accuracy (PA)** | [0.76, 0.94] | [0.89, 0.86] | [0.79, 0.85] | [0.93, 0.94] | [0.94, 0.88] | [0.89, 0.86] | [0.97, 0.79] |
| **Overall Accuracy (OA)** | 0.88 | 0.84 | 0.87 | 0.93 | 0.93 | 0.88 | 0.93 |
| **Kappa Coefficient (KC)** | 0.78 | 0.74 | 0.83 | 0.87 | 0.82 | 0.74 | 0.79 |
| **F-score** | [0.88, 0.75] | [0.90, 0.90] | [0.90, 0.89] | [0.94, 0.93] | [0.95, 0.87] | [0.90, 0.84] | [0.96, 0.83] |
| **Mapping Result Area (km$^2$)** | 13,298.61 | 21,230.76 | 8963.92 | 23,793.60 | 26,755.34 | 5035.26 | 7285.36 |
| **Reference Area (km$^2$)** | 14761 | 20192 | 9400 | 22192 | 25121 | 5314 | 8075 |
| **Relative Error (%)** | −9.91 | 5.14 | −4.64 | 7.22 | 6.51 | −5.25 | −9.78 |

## 4. Discussion

### 4.1. Comparative Analysis of Zoning-Mapping Results

As can be observed in the graphs and tables in the Results section, the results of all three accuracy evaluation methods have indicated that the accuracy of rice mapping in the three northeastern provinces was better than that in the seven southern provinces. After a long-term survey of satellite high-resolution images, it was determined that the areas to the south and north of the Q-H Line varied in terms of topography, phenology, and agricultural activities to different degrees, leading to remarkable discrepancies in the adopted mapping methods and mapping effects. These differences are manifested in the following three aspects.

In terms of topography, the distribution of paddy fields in the lower terrain in the north was more regular and coherent than that in the south since there were fewer hilly mountains, which eased visual interpretation, and the samples were more reliable. On the contrary, the distribution of rice fields in several provinces in the southern region was more fragmented and hybrid in nature, and the mixing effect of pixels was prominent, which made the visual interpretation of rice sample points harder while rendering the samples less stable.

In terms of phenology, the three northeastern provinces are located in the middle temperate zone, and the rice maturation system is mono-annual; therefore, their phenological characteristics are obvious, and it is convenient to interpret the four key phenological stages from the spectral profiles of rice [81]. The local rice maturation system was found to be rela-

tively diverse due to the water and heat conditions in the seven southern provinces. There was a mixture of mono-annual, bi-annual, and even tri-annual maturation, which resulted in flexible phenological characteristics and made it difficult to determine the reasonable boundaries of the phenological periods.

In terms of farming activities, the relatively simple rice maturation scheme in the northern region has allowed for only a single-phase planting process, making the inter-annual rice field distribution changes more stable and the samples more consistent over time. Even if the year of mapping and the year of reference image did not match, the actual location of the sample did not change drastically. In contrast, under the complex rice maturation scheme in the southern region, farmers tended to adopt complex planting measures such as crop rotation and intercropping [82], which led to the difficulty in visual the interpretation of the rice sample points and the lack of distinct V-shaped characteristics of the microwave sequences.

In addition, the identified rice fields were usually accompanied by a certain number of tiny holes within them, resulting in a slightly underestimated extracted area compared to the official statistical area in the rice-mapping results of the provinces in the northern region. Many irregular patches were observed in the extraction results of the southern region. This could be the result of the noise problem in the Sentinel series data source itself and the adopted pixel-by-pixel classification method's failure to account for neighborhood information mining [83,84]. Both of these problems could be addressed by implementing morphological methods to filter out spurious pixels or repair voids, but there has not yet been a scientific solution that can completely solve them [84–86].

### 4.2. Features and Uncertainties of This Study

In response to the need for increasingly efficient and fine-grained arable land resource survey and management methods and considering the lack of national-scale rice mapping research, a rational and innovative zoning strategy based on the Q-H Line was proposed in this study in view of the geospatial heterogeneity of rice distribution and the corresponding phenology nationwide. On this basis, a high-resolution rapid mapping of mid-season rice in the entire monsoon region of China was conducted on the GEE cloud-computing platform, which possesses a massive amount of remote-sensing data storage and a built-in machine learning classifier. Furthermore, the large-scale mapping of rice in a short period of time under the condition of a relative scarcity of ground truth samples was achieved by minimizing the influence of cloud pollution, leading to the acquisition of encouraging results, which could provide valuable references for similar studies and applications in the future. In addition, the extension of the mapping years by adjusting the time span parameters was also achieved, which could provide a possible method of producing long time series and large-scale high-resolution rice datasets.

However, there are some obvious shortcomings in this study that should be overcome in future in-depth studies. Firstly, regarding the choice between accuracy and efficiency, which is often faced in remote-sensing image classification, efficiency was favored in this study, resulting in a lack of quantity and quality of the selected sample points. This limitation could have resulted in unstable mapping results and accuracy evaluation. Secondly, the classifiers adopted might be the most basic machine learning classifiers that have been applied for a long time, and the parameters of the algorithms might be too roughly designed and not sufficiently adapted to the local conditions. Thirdly, the identification of rice phenological stages might not have been precise enough, as the entire life cycle of rice was only divided into four stages in the northern region, while a simple statistically based phenological calendar was adopted in the southern region, which could also have negatively affected extraction accuracy. Fourthly, in the preprocessing steps, pixels with slopes greater than 15 degrees were removed in order to exclude terrains where rice may not be planted to enhance the efficiency of RF, while some of the paddy rice growing in the terraces were inevitably omitted, thus influencing the mapping accuracy. Moreover, the insufficient number of effective images, the low temporal resolution of the Sentinel-1 data,

the inherent border noise of the Sentinel-1/2 data, and the single rice species selected for extraction would also deserve great attention.

Finally, another crucial factor of uncertainty affecting the mapping accuracy is the size of the buffer zone (i.e., the value of N) of the paddy field samples. In the Methodology section, it was stated that N needs to be initially adjusted according to the actual average size of the paddy fields in each region (whose optimal value can only be determined through undertaking several tests). However, in practice, this process may be time-consuming and require a combination of field research and empirical knowledge. If the selected N value is larger than the average size of the local paddy fields, the samples input to the classifier will be mixed with features of other land use types, and the sample purity will be reduced, resulting in a greater predicted area due to the misidentification of non-paddy fields. If the selected N value is smaller than the average size of local paddy fields, the sample features learned by the classifier will be insufficient, and the feature heterogeneity within the paddy fields will be ignored; thus, part of the paddy fields will be ignored, some paddy fields are not fully included in the classification results, resulting in a smaller predicted area. Therefore, in future studies, it's suggested to develop an adaptive buffer size selection method, i.e., to find the optimal N values for various areas using a cyclic feedback tuning method.

### 4.3. Future Research Directions

In response to the shortcomings mentioned above and the common shortcomings of similar studies, futural exploratory research might be conducted around the following four aspects.

- **Fusion of multi-source data.** In the future, researchers could try to fuse multi-source remote-sensing data (Sentinel-1/2, Landsat, MODIS, UAV, etc.) to improve the feature index system and enhance classification effects. Some experimental studies have been conducted, for which more reliable results have been obtained, which could provide new insights for mapping crop cultivation spatial distribution [87–90].
- **Synergy of cloud computing and deep learning**. The deep learning algorithms that have emerged in the last decade have improved rice extraction on complex surfaces and in fragmented landscapes by building a moderate number of neuronal computation nodes and multi-layer operational hierarchies with higher classification accuracy compared to traditional machine learning algorithms [91,92]. However, more complex model structures would also require better hardware performance, longer training times, and a larger number of data labels [93]. Therefore, researchers could attempt to combine the advantages of the high accuracy of deep learning and the high efficiency of the GEE platform to construct local deep learning models on this cloud-computing platform.
- **Enhancement of post-processing**. For the optimization of the results after GEE derivation, other morphological methods could be considered besides performing majority filtering; different processing window sizes (dynamic windows) could also be experimented with to further filter out noise and repair voids, thus compensating for the deficiencies of the underlying data and algorithms [94,95].
- **Development of sample-poor mapping technologies**. A lack of accurate sample points might become the norm for future large-scale crop mapping efforts and a bottleneck for technological progress in related fields [76,96,97]. Therefore, the development of deep learning classification algorithms with stronger autonomous learning capacity via coupling deep learning frameworks (e.g., Tensor Flow) or large artificial intelligence models (e.g., the ChatGPT) with emerging technologies would be an important way in which to improve the accuracy of large-scale rice extraction models and their generalization capability [98–100].

## 5. Conclusions

In this study, a national-scale mid-season rice zonal mapping strategy was proposed under the conditions of the relative scarcity of ground-truthing sample data given the spatial heterogeneity of rice phenology across China. Secondly, based on Sentinel-1/2 data and machine learning algorithms, a 2021 Chinese monsoon zone mid-season rice cultivation extent dataset was created on the GEE platform, which contains maps of mid-season rice fields for 28 provincial administrative regions in China. The data file format is GeoTIFF, with a spatial resolution of 10 m, and the geo-reference system used was WGS84 (EPSG:4326). Finally, with the help of multiple methods, the accuracy evaluation results have shown that the relative errors of the mapped areas of mid-season rice in each province were limited to 10% or less compared with the statistical yearbook, while the overall accuracy obtained from the confusion matrix exceeded 85%. Restricted by the availability of samples and geographical complexity, the accuracy of mapping in the south turned out to be generally lower than that in the north. Additionally, the extension of the mapping years by adjusting the time period parameters can be achieved. The inspiring results obtained in this study provide a solution that can be referenced to produce long-time-series, large-scale and high-resolution spatial datasets of rice cultivation extent. In the future, efforts could be made toward the fusion of multi-source remote-sensing data, the fusion of cloud-computing platforms and deep learning models, the reinforcement of the post-processing of extraction results, and the development of intelligent mapping methods in the absence of abundant samples in order to realize further synergistic progress in accuracy and efficiency in research and applications related to large-scale crop mapping.

**Supplementary Materials:** The following supporting information can be downloaded at: https://www.mdpi.com/article/10.3390/rs15164055/s1, Table S1: Allocation of rice selection points by provincial administrative units within the whole China monsoon region.; Table S2: Formula for calculating classification coefficients based on confusion matrix.

**Author Contributions:** Conceptualization, C.H. and J.D.; methodology, C.H.; data curation, C.H. and J.Z.; investigation, J.Z.; validation, C.H.; writing—original draft preparation, C.H.; writing—review and editing, J.D. and P.L.; visualization, C.H.; supervision, S.Y. and A.L.; project administration, J.D.; funding acquisition, J.D. All authors have read and agreed to the published version of the manuscript.

**Funding:** This research was funded by the Outsourcing Project of the Center for Remote Sensing Application of Land and Satellite, Ministry of Natural Resources, the Consulting Project of Chinese Academy of Engineering (2023-30-13), and the National Key Research and Development Program of China (2020YFC1807501).

**Data Availability Statement:** The data presented in this study are available on request from the corresponding author. The data are not publicly available due to projects that funded this study has not yet been fully recognized as complete, thus premature disclosure of the data would potentially create a privacy or moral hazard.

**Conflicts of Interest:** The authors declare no conflict of interest.

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
