# Peer review of "High-Resolution National-Scale Mapping of Paddy Rice Based on Sentinel-1/2 Data"

_remotesensing, doi:10.3390/rs15164055_

Round 1

Reviewer 1 Report

The authors demonstrate that GEE gives a feasible implementation solution for the rapid identification of rice cultivation across China. GEE could provide a solution for rapid identification using optical and microwave data. 

Considering geographical heterogeneity, the authors have managed to obtain good accuracy in their classification. The strategy for zoning, data selection, selection of parameters, and its implementation using various algorithms have been implemented with relatively good success considering the spatial extent of the study area.

A map depicting the national mid-season rice production area in 2021 with a spatial resolution of 10 meters was developed. Such analysis encompassing the monsoon region of China can be a great tool for government and non-government agencies to manage rice yield, production, processing, storage, and distribution effectively. As rice is a staple food and meets a major proportion of the daily calorie intake of millions of people, such studies can provide a framework to map rice for nationwide and global studies. The classification accuracy can be improved further. The article demonstrates the effective use of remote sensing in covering large areas to solve pertinent issues. Recommends publication of the article in the journal. 

Author Response

Dear Reviewer,

    We deeply appreciate the time and effort you have taken to evaluate our paper. We are also very grateful to receive your approval. In order to further improve the soundness of this study and the readability of the paper, we have made the necessary extensive revisions to various aspects of the paper such as methodology, results, and language. We have highlighted the revised sections in the manuscript that address the comments and invite you to review them again.

    Thanks again and best wishes!

Reviewer 2 Report

Mapping national-level paddy rice fields is hampered by heterogeneous rice phenology. The contribution of this study is the usage of a zoning strategy to divide study region into the north and the south, where different mapping algorithms are applied. The study is well designed, and I enjoy reading it.

Here are some concerns:

1.       Line 58: It’s better to replace “rice grow rhythms” with “rice phenology”.

2.       Line 73-75: Beyond the optics-based mapping and the microwave-based mapping, the combination of optics and microwave is an important class of rice mapping algorithms.

3.       Line 120:  Reviews on large-scale rice mapping studies are missing in the introduction. There are some national rice maps, for example:

Ruoque Shen, Baihong Pan, Qiongyan Peng, et al. High-resolution distribution maps of single-season rice in China from 2017 to 2022[DS/OL]. V3. Science Data Bank, 2023[2023-05-31]. https://doi.org/10.57760/sciencedb.06963. DOI:10.57760/sciencedb.06963.

4.       Line 236: Exclusion of area with slopes greater than 15 degree risks omitting paddy rice fields in terrace. You should state this limitation.

5.       Line 279: Replace “Random segmentation” with “random sampling”.

6.       Line 465: What are these default values?

7.       Line 478: I suggest you add a comparison with existing rice map as a way of validation.

8.       Line 494: Where is the result of statistic yearbook comparison? It is missing in Result or Discussion section.

9.       Line 648: The order of the three adjectives are not reasonable. It should be “a mixture of mono-annual, bi-annual, and even tri-annual …”.

Reviewer 3 Report

Dear Authors,

I would like to express my appreciation for your interesting paper titled "High-resolution National-scale Mapping of Paddy Rice Based on Sentinel-1/2 Data." It is evident that your research holds significance within the field. However, I have some comments that I believe could further enhance the value of your work.

Firstly, I recommend considering a reduction in the initial part of the paper where you explain the production of rice in China and provide an overview of Copernicus data. While these details are widely known, I suggest focusing on specific aspects that are critical for the overall accuracy of the mapping results. For instance, discussing the normal size of rice fields would be valuable, as it can have an impact on the accuracy of the mapping. It would be beneficial to elaborate on how field size affects the accuracy and discuss any implications arising from this relationship.

Secondly, I encourage you to provide a more detailed explanation of how the data from Sentinel-1 and Sentinel-2 were combined. Based on our experiences, combining data from these two sources can sometimes result in decreased accuracy. Therefore, it would be valuable to explore the reasons behind this and address any challenges or limitations encountered during the data integration process. By providing additional insights into the methodology, readers will gain a better understanding of the data fusion techniques employed.

Additionally, the influence of cloud cover on Sentinel-2 analysis is not mentioned in the paper. Considering the common challenge of cloud cover in remote sensing, it would be important to acknowledge its impact on the accuracy and reliability of the mapping results. I recommend discussing how you addressed the issue of cloud cover during the analysis, as this will contribute to the scientific rigor of your study.

In conclusion, by incorporating these suggestions, I believe you can further enhance the value of your paper for the scientific community. Focusing on critical details such as field size, addressing challenges in data integration, and discussing the influence of cloud cover will provide comprehensive insights and contribute to the existing knowledge in the field.

Thank you once again for your valuable contribution to the field. I appreciate your efforts and look forward to seeing further advancements in your research

Round 2

Reviewer 3 Report

Dear authors,

thanks for update and eplanation, I think, that paper can be published